# Incomplete Exhalation during Resuscitation—Theoretical Review and Examples from Ventilation of Newborn Term Infants

**DOI:** 10.3390/children10071118

**Published:** 2023-06-28

**Authors:** Thomas Drevhammar, Peder Aleksander Bjorland, Joanna Haynes, Joar Eilevstjønn, Murray Hinder, Mark Tracy, Siren Irene Rettedal, Hege Langli Ersdal

**Affiliations:** 1Department of Women’s and Children’s Health, Karolinska Institutet, 171 77 Stockholm, Sweden; 2Department of Paediatrics, Stavanger University Hospital, 4019 Stavanger, Norway; 3Department of Anaesthesia, Stavanger University Hospital, 4019 Stavanger, Norway; 4Faculty of Health Sciences, University of Stavanger, 4021 Stavanger, Norway; 5Laerdal Medical, Strategic Research Department, 4007 Stavanger, Norway; 6Department of Paediatrics and Child Health, Sydney University, Westmead, Sydney, NSW 2006, Australia; 7Neonatal Intensive Care Unit, Westmead Hospital, Westmead, Sydney, NSW 2145, Australia

**Keywords:** resuscitation, infant, newborn, positive end-expiratory pressure, positive pressure ventilation, intrinsic

## Abstract

Background: Newborn resuscitation guidelines recommend positive pressure ventilation (PPV) for newborns who do not establish effective spontaneous breathing after birth. T-piece resuscitator systems are commonly used in high-resource settings and can additionally provide positive end-expiratory pressure (PEEP). Short expiratory time, high resistance, rapid dynamic changes in lung compliance and large tidal volumes increase the possibility of incomplete exhalation. Previous publications indicate that this may occur during newborn resuscitation. Our aim was to study examples of incomplete exhalations in term newborn resuscitation and discuss these against the theoretical background. Methods: Examples of flow and pressure data from respiratory function monitors (RFM) were selected from 129 term newborns who received PPV using a T-piece resuscitator. RFM data were not presented to the user during resuscitation. Results: Examples of incomplete exhalation with higher-than-set PEEP-levels were present in the recordings with visual correlation to factors affecting time needed to complete exhalation. Conclusions: Incomplete exhalation and the relationship to expiratory time constants have been well described theoretically. We documented examples of incomplete exhalations with increased PEEP-levels during resuscitation of term newborns. We conclude that RFM data from resuscitations can be reviewed for this purpose and that incomplete exhalations should be further explored, as the clinical benefit or risk of harm are not known.

## 1. Introduction

Positive pressure ventilation (PPV) is needed by approximately 5% of term newborn infants [1]. Establishing effective spontaneous or mechanical ventilation is critical for transition after birth. Management of ventilation is extensively discussed in resuscitation guidelines [1,2,3], but ventilation targets, methods of providing PPV, and how to monitor ventilation delivery have been insufficiently studied [4]. Examples of uncertainties in management are the potential benefits of real-time feedback using more advanced monitors such as respiratory function monitors (RFMs), and the delivery of PEEP (level and device used). RFMs provide data that can be used to optimise PPV, for example by confirming ventilation and avoiding leakage. RFMs can also perform more complex measurements of respiratory function and dynamic changes during resuscitation.

Ventilation of newborn infants is mainly focused on lung inflation, and guidance on exhalation is sparse [2]. Research into the properties of exhalation has been given relatively little attention, apart from the use of PEEP and continuous positive airway pressure (CPAP).

A wide array of devices and interfaces can be used for resuscitation and stabilisation. Commonly used systems for providing PPV during newborn resuscitation include T-piece resuscitators and self-inflating bags, with a face mask, laryngeal mask airway (LMA) or endotracheal tube as the interface between device and neonate. T-piece systems have become popular due to the ability to deliver CPAP in the breathing newborn with respiratory distress and PPV for the apnoeic infant. These systems have common features of a constant inflow, a resistor (the PEEP valve) to provide CPAP and a flow exit port in the PEEP valve which, when occluded, provides rise in pressure to the pre-set desired peak inflation pressure (PIP). T-piece systems with PEEP are often used in term infants but there have been no clear recommendations for the use in this population [2]. T-piece systems have high imposed resistance to spontaneous breathing but also to expiratory flows during PPV [5,6]. This can be explained by the resistor valve that is used to adjust and generate PEEP or CPAP [7,8]. T-piece resistance has the potential to prolong exhalation because it increases the expiratory time constant, particularly at the lower recommended fresh gas flow rate of 8LPM [8,9]. Investigations of different resuscitation device properties, the PEEP level and clinical outcomes are complex because it is difficult to isolate insufflation from exhalation. This is illustrated, for example, by fresh gas flow determining both inspiratory rise time and expiratory resistance in T-piece systems.

Incomplete exhalations have been described and debated for more than 40 years and several terms have been used [10]. The terms used either refer to a PEEP higher than desired (e.g., auto, intrinsic and inadvertent-PEEP) or to a volume not exhaled (e.g., air trapping and hyperinflation). Incomplete exhalation can occur during both PPV and spontaneous breathing [11]. It can be present as a part of normal physiology in several mammals and is common in newborn infants [12,13]. Incomplete exhalation and gas trapping has been described in previous studies using RFMs [14] but is mainly recognised in intubated, mechanically ventilated, infants [15].

The time needed for complete exhalation is given by the expiratory time constant and has been well described [16,17,18]. This is the product of expiratory resistance (R_e_) and respiratory system compliance (C_rs_). The function is exponential and 95% of the volume will be exhaled after three time constants. The mechanical theory appears simple at first, but non-linearity and inhomogeneous compliance and resistance add complexity [19].

RFMs have been used both as an intervention to improve care and to investigate aspects of newborn resuscitation and stabilisation. Currently, RFMs are not recommended in guidelines or reviews as a tool to optimise clinical outcomes, reduce mask leakage and/or achieve specific ventilation targets, due to limited evidence [20,21]. However, RFMs are essential as research tools because data on compliance, resistance and ventilation cannot be obtained without measuring flow and pressure. Leakage is of particular concern when analysing data from RFMs and it is difficult to estimate the effects on expiratory flows and pressures. In manikins the majority of leakage during PPV was in the inspiratory part of the breathing cycle [22]. Using RFMs to optimise PEEP and exhalation in relation to properties of devices has not been attempted as far as we know. A particular clinical challenge will be the dynamic changes in lung physiology and the urgency of managing a non-breathing infant.

Our aim was to study examples of incomplete exhalations in term newborn resuscitation and discuss these against the theoretical background.

## 2. Materials and Methods

Examples of flow and pressure patterns were extracted from a study at Stavanger University Hospital conducted from 1 June 2019 to 31 March 2021. Details are provided in the original publication [23]. Informed consent was obtained from all participants and the study was reviewed by the regional ethical committee (REK vest 2018/338).

Resuscitation was undertaken according to national standards. PPV was provided with T-piece systems (NeoPuff, Fisher&Paykel Healthcare, Auckland, New Zealand) at PIP 30 cm H_2_O and PEEP 5 cm H_2_O. The fresh gas flow was visually set to 8 L/min. In the selected examples there was no adjustment of fresh gas flows. Inspiratory oxygen concentration was adjusted with a blender and did not affect fresh gas flow. The recorded pressures and flows were not presented to the resuscitation team. In total, 129 term infants who received PPV were reviewed to provide the selected examples.

The Laerdal Newborn Resuscitation Monitor (Laerdal Global Health, Stavanger, Norway) has been used in other studies [23,24,25]. Pressure was measured at the interface using a standard sensor (MPXV5010, Freescale Semiconductor Inc., Austin, TX, USA) and flow with a hotwire technique (MIM Gmbh, Krugzell, Germany). The resuscitations were video recorded and patient data extracted from medical records. The time of birth was logged by the midwife assistant in the Liveborn application (Laerdal Global Health, Stavanger, Norway) and time from birth to start of PPV is presented in Table 1. The first inflation was used as time zero in the figures.

**Table 1 children-10-01118-t001:** Description of incomplete exhalation examples in figures.

	Type	History	PEEP in Examples (mbar)
Figure 1	Short expiratory time	Female 3634 g, 41 + 3 wGA. Vaginal delivery and not breathing. Apgar 1/5/5. PPV started at 58 s. Stable breathing at 16 min.	6.9, 5.3 *
Figure 2	Increased tidal volume	Male 3444 g, 37 + 6 wGA. Instrumental delivery (vacuum) and not breathing. PPV started at 38 s. Apgar 6/9/10. Stable breathing at 2 min.	5.6, 6.5, 7.9
Figure 3	Increased tidal volume	Male 3740 g, 40 + 2 wGA. Vaginal delivery and not breathing. Apgar 2/1/5. PPV started at 75 s. Stable breathing at 12 min.	4.1 *, 4.3 *, 4.0 * (90 s) 6.8, 6.1, 6.6 (183 s)
Figure 4	Increased resistance Infant Intubated	Male 4030 g, 40 + 0 wGA. Instrumental delivery (vacuum) and not breathing. Apgar 2/2/5. PPV started at 148 s. Intubated at 5 min.	4.9 *, 5.1 *, 4.7 * (no ET) 6.1, 5.6, 5.9 (ET)
Figure 5	Increased resistance	Male 3560 g, 40 + 2 wGA. Instrumental delivery (forceps) and not breathing. Apgar 3/6/9. PPV started at 78 s. Stable breathing at 4 min.	7.3, 6.7, 6.2

The PEEP level reported in chronological order of breaths and (*) indicates breaths with complete exhalation and no or very low increase in PEEP. Abbreviations in table: wGA, weeks gestation age and ET, endotracheal tube.

The recorded pressure, volume and flow traces were visually inspected for examples of incomplete exhalation. These were identified as flow not returning to zero with a PEEP higher than intended. We used no predetermined limits or cut-off values for end expiratory flows or PEEP for selection. Only recordings of high quality with no artefacts and minimal leakage were reviewed. Leakage was calculated as the difference between inspiratory and expiratory tidal volume and was below 20% in the selected examples. Leakage mainly occurs during insufflation and the selection of breaths with minimal leakage was undertaken to exclude leakage during expiratory flows. In a mannikin study of PPV there was no leakage during exhalation at leakages below 50% [22]. The volume panels in the examples were calculated from the recorded flow and zeroed after each breath. For the selected examples, with minimal leakage, the expiratory and inspiratory volumes are similar.

Review of recordings was performed using Matlab R2022b (MathWorks, Natick, MA, USA). No statistical analyses were performed in the manuscript, and resistance, compliance and expiratory time constants were not calculated or reported.

## 3. Results

Five examples of term infants with incomplete exhalations were selected (Table 1). The levels of PEEP, measured at the face mask, in the examples of incomplete exhalation breath range from 5.6 to 7.9 mbar (1.00 mbar equal to 1.02 cm H_2_O). The first three figures illustrate the importance of tidal volume and expiratory time. Figure 4 and Figure 5 represent increased resistance. The five examples were selected as illustrations of lung mechanics during PPV and to introduce factors affecting the time needed to complete exhalation.

The first example demonstrates short expiratory time and flow not returning to zero (Figure 1). This corresponded to a minor increase in PEEP. In this example, inspiratory–expiratory ratio (I:E) was inversed with inspiratory time being longer than expiratory time. With larger tidal volumes, additional time for exhalation is needed (Figure 2). The larger tidal volume may be related to increased compliance during the first minutes after birth (Figure 3).

An example of increased resistance from an endotracheal tube is provided in Figure 4. This is related to the increased resistance from the endotracheal tube and a reduction in peak expiratory flows. The tidal volume was reduced and the exhalation was incomplete. An increase in both inspiratory and expiratory resistance was seen in Figure 5. In this infant, neither inhalation nor exhalation were complete. The reason for the increased resistance is not known.

## 4. Discussion

We found examples of incomplete exhalations and low levels of increased PEEP at the face mask when reviewing previously recorded RFM data from term infants. The examples illustrate that the theoretical basis of exhalation also applies to term infant resuscitation and that incomplete exhalations are present in clinical recordings.

During PPV, incomplete exhalation is related to an early start of the next inflation, the volume exhaled and the total resistance [11]. The decline in pressure from PIP to PEEP is an exponential function (e.g., Figure 1) and is described using expiratory time constant. Time constants (τ) are the products of resistance and compliance (τ = R_e_ × C_rs_) and 1 τ defines the time taken for pressure to decline by 63%. During PPV, compliance is directly related to tidal volume (TV) and driving pressure (C_Rrs_ = TV/(PIP − PEEP). The inhomogeneous lung has several time constants (regional differences in compliance and resistance), and both compliance and resistance are non-linear. These complex non-linear relationships and exponential functions are often simplified to allow a clinically relevant understanding [19].

### 4.1. Expiratory Time

Starting a new insufflation before expiratory flows have returned to zero will lead to an incomplete exhalation (Figure 1). Examples of shortening the expiratory time include increased ventilation rate or increased inspiratory–expiratory (I:E) ratio. The standard method by which to describe how flow and pressure decrease over time is the expiratory time constant. Time constants limit ventilation frequencies and minute ventilation that can be used without adjusting pressures or generating intrinsic PEEP [10,11].

### 4.2. Tidal Volume and Compliance

Expiratory time constants during PPV are directly correlated to compliance and TV. An increase in tidal volume can contribute to, or explain, incomplete exhalations (Figure 2 and Figure 3). During transition, the gradual improvement in compliance should increase the likelihood of incomplete exhalation [4]. Another example is the increased compliance seen after surfactant administration [18].

During resuscitation, the large variation in delivered tidal volume has been explained not only by change in compliance, but also leakage, obstruction, and spontaneous breathing efforts [23,26].

### 4.3. Resistance

Expiratory time constants are directly correlated to total resistance. The total resistance in the examples is a combination of imposed resistance from the ventilation device and the infant’s airways. The device–interface resistance of a standard face mask is very low, but an endotracheal tube caused increased resistance, as shown in Figure 4. In this example the resistance increased, and the tidal volume decreased after intubation. Even with this volume decrease, incomplete exhalation was seen. In Figure 5 the cause for increased resistance was not known but affected both inspiratory and expiratory flows.

### 4.4. Respiratory Function Monitors and Limitations

The role of RFMs in training for, and clinical management of, resuscitation has not been established but their use has been commented on in guidelines and discussed in reviews [20,21]. The examples used in this manuscript could not have been obtained by means other than the use of RFMs.

A limitation to the examples is that they only represent a small number of high-quality recordings. The majority of breaths were unsuitable due to leakage, artefacts or because there were no signs of incomplete exhalation. Analysis of the complete dataset to describe the frequency of artefacts, leakage and frequency of incomplete exhalations requires development of software and methods. The recordings were also complex, reflecting changes in lung mechanics over time and clinical responses to resuscitation. A strength of the study was that the user was blinded to the data collection from the RFM. It is possible that higher quality data without leakage could have been obtained if the team had had access to real-time RFM data.

In a recent review, RFM use did not conclusively show reduced leakage or achievement of desired tidal volumes [21]. Reducing leakage should increase the risk of incomplete exhalation from higher imposed resistance, increased PEEP and larger tidal volumes. It is possible that displaying flow curves could balance this by clinical identification of incomplete exhalations, but the visual inspection and interpretation of these data have been identified as difficult even in intubated patients on mechanical ventilation [27].

The pressures in the examples were measured between the T-piece and patient interface. Because exhalation is not complete, the true PEEP in the lungs, i.e., the intrinsic PEEP, will be higher than that measured at the interface. Investigating intrinsic PEEP during resuscitation by using expiratory pauses was not an option. The increased PEEP at the face mask reported in Table 1 is a direct consequence of the T-piece resistance and the flow not returning to zero. Using end-expiratory recorded pressure at the face mask as a variable that reflects incomplete exhalation has several limitations. It will only work in systems with high resistance and will not work in a system with lower resistance. For T-piece systems this pressure will be affected by use of higher fresh gas flows and higher PEEP. This also limits the possibility of comparing PEEP at the face mask delivered with other systems than that used to generate our data. For example, mask pressure data from self-inflating bags or ventilators have to be interpreted with caution. The T-piece settings used for all patients included in the study employed the same, unchanged, level of PEEP and fresh gas flow, allowing us to compare PEEP at the face mask instead of relying only on end-expiratory flow.

### 4.5. Clinical Importance

The clinical importance of incomplete exhalations is not known. Our examples are from term infants, however T-piece resuscitators are also commonly used both in preterm- and very low birth weight infants. An increase in compliance might be balanced by slower ventilatory rate. Any increase in resistance (e.g., endotracheal tube, T-piece set to high CPAP with low fresh gas flows) will increase the risk [6,8].

A potential risk of delivering higher-than-intended pressure to the lungs is increased risk of air-leaks and pneumothoraces. Three observational studies have found an increasing risk when CPAP and PEEP were used routinely [28,29,30]. Incomplete exhalation could be a component, or contributing factor, not previously considered. Testing this hypothesis will be challenging, given that air-leaks and pneumothoraces are not common. We strongly believe that such investigation should be encouraged.

A different hypothesis is that incomplete exhalations could be beneficial and a gentle way to build FRC. This is a theoretical alternative to the use of a higher base-line PEEP. It can be achieved by intentional use of resistance from devices and interfaces (increasing expiratory time constants) or the use of higher ventilation rates, including inverse ratio ventilation (shorter expiratory times). Intentional use of intrinsic PEEP has the disadvantage of being difficult to measure and there should be concerns about delivering inadequate ventilation or high airway pressures.

Clinical use of RFMs to guide resuscitation was not investigated in the clinical trial and the resuscitation team was blinded to the RFM data. Whether RFMs can be used during resuscitation to prevent or detect incomplete exhalations, inadvertent PEEP, or air leaks and pneumothoraces, is an open question.

There are clinical challenges in providing adequate ventilation without using too high pressures. Gentle pressures and lower volumes may reduce the risk of barotrauma, volutrauma and lung injury, but increase the risk of hypoventilation and reduced aeration [4]. Incomplete exhalations during PPV have not previously been discussed in this context. Optimising delivery of PPV should continue to be the focus of resuscitation research and guidelines.

## 5. Conclusions

The physiology and mechanical explanations of incomplete exhalation have been well described and examples during resuscitation of newborn infants were present in our data. We encourage further research on exhalations during resuscitation and review of completed trials using RFM data. This should add new knowledge on incomplete exhalations and clinical benefit or potential harm. Potential causes, such as reduced time for exhalation, high respiratory rate, inverse I:E ratios, large tidal volumes and high imposed resistance, should be explored. Incomplete exhalation could be a variable not previously considered when assessing factors related to quality of resuscitation and PEEP.

## Figures and Tables

**Figure 1 children-10-01118-f001:**
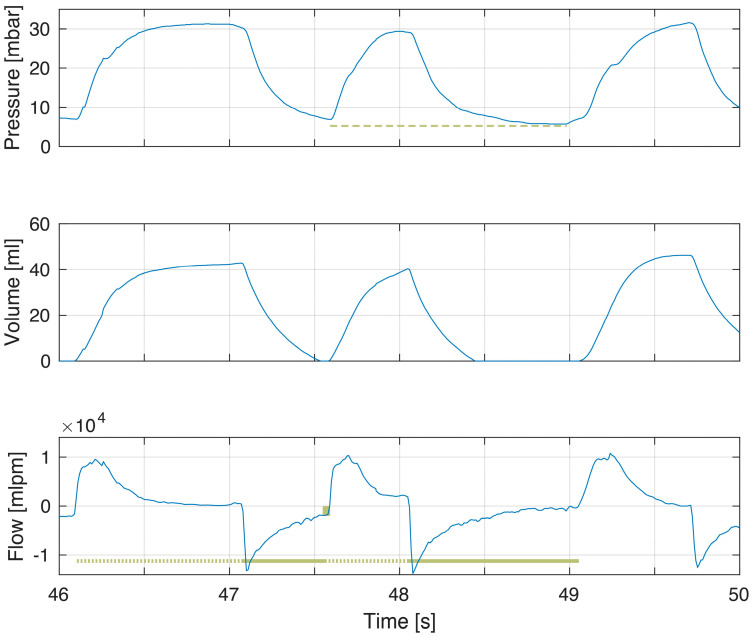
Example of a breath with incomplete exhalation. The expiratory time of the first breath was shorter than for the second breath (flow panel solid line) and the expiratory flow does not return to zero (flow panel box indicating magnitude of continued expiratory flow at point of gas flow reversal to inflation). There was higher PEEP with incomplete exhalation (pressure panel dashed line). PEEP was 6.9 cm H_2_O in first breath and 5.3 cm H_2_O in second breath.

**Figure 2 children-10-01118-f002:**
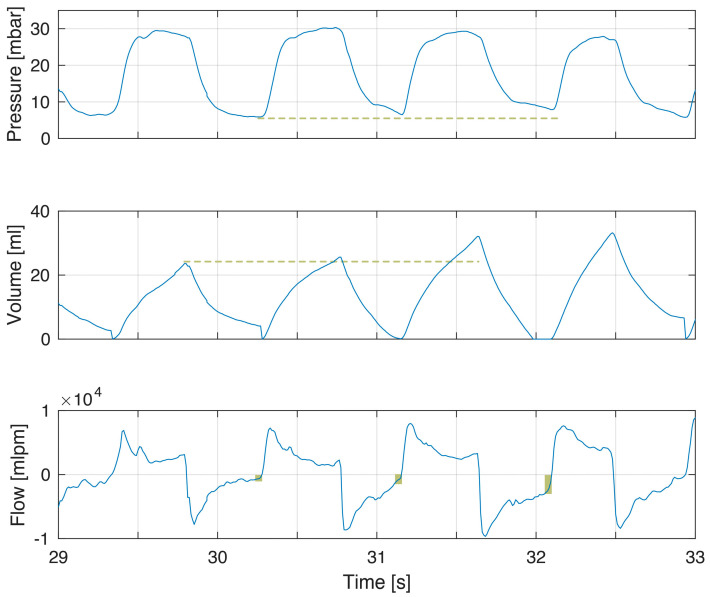
Example of incomplete exhalation related to increased tidal volumes. The modest increase in tidal volumes (volume panel dashed line) gives incomplete exhalations of increasing magnitude (flow panel box indicating magnitude of continued expiratory flow at point of gas flow reversal to inflation) and increased PEEP (pressure panel dashed line). PEEP was 5.6, 6.5 and 7.9 cm H_2_O in the three breaths.

**Figure 3 children-10-01118-f003:**
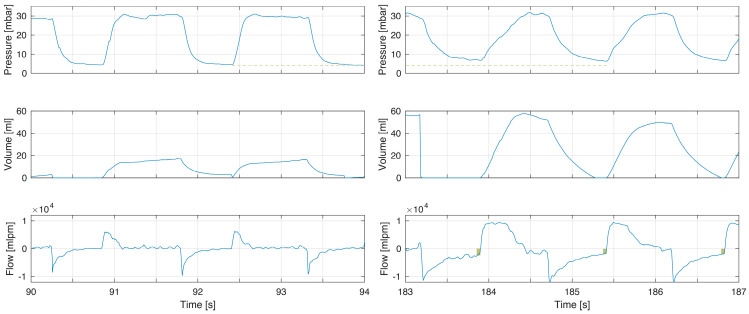
Example of increased compliance and tidal volumes during the first minutes of life. The increase in tidal volume in this infant led to incomplete exhalations (right flow panel boxes indicating magnitude of continued expiratory flow at point of gas flow reversal to inflation) with increased PEEP (pressure panel dashed line comparing left to right). This can be explained by the dramatic increase in compliance. The longer time to reach PIP is a consequence of fresh gas flow rate when using TPR as lung compliance increased (right pressure panel).

**Figure 4 children-10-01118-f004:**
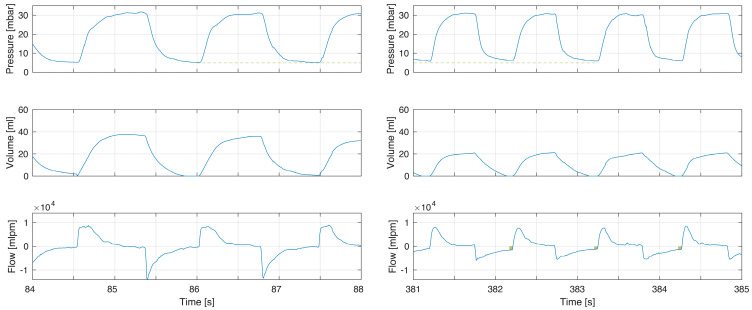
Example of increased resistance after intubation. Before intubation (left) there was no incomplete exhalation and flow returned to zero. After intubation the increase resistance led to incomplete exhalations (right flow panel box indicating magnitude of continued expiratory flow at point of gas flow reversal to inflation) with an increased PEEP (pressure panel dashed line comparing left to right). After intubation there was reduced compliance and respiratory rate was increased.

**Figure 5 children-10-01118-f005:**
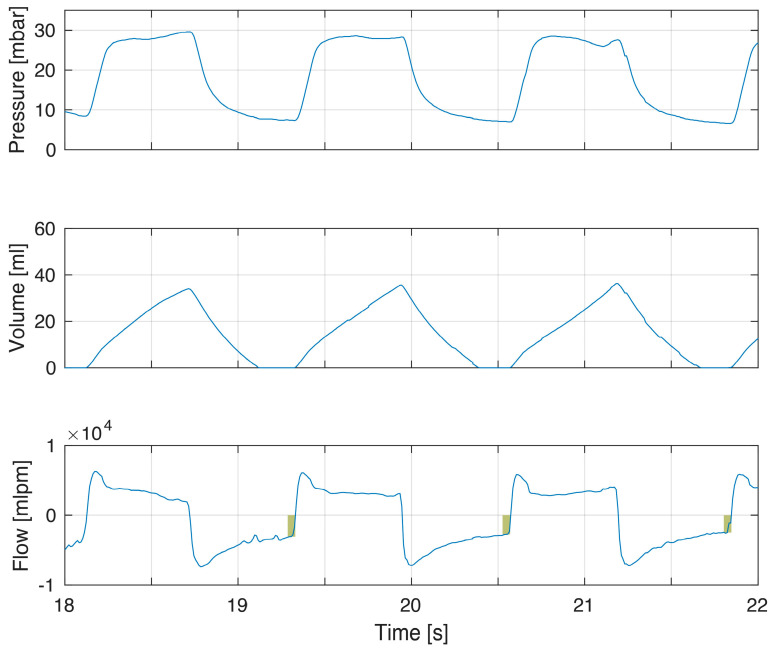
Example of increased resistance on both inspiration and expiration. This infant showed incomplete exhalations (flow panel box indicating magnitude of continued expiratory flow at point of gas flow reversal to inflation) but also incomplete inflations with flow not returning to zero and increasing tidal volume when insufflation was terminated. The reduced rate of inspiratory flows can partly be explained by high compliance, but flows are also clearly below the set fresh gas flows, indicating resistance as a cause.

## Data Availability

Original data are available upon request. Contact the corresponding author.

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
