# Peer review of "Incomplete Exhalation during Resuscitation—Theoretical Review and Examples from Ventilation of Newborn Term Infants"

_children, 2023, doi:10.3390/children10071118_

Round 1

Reviewer 1 Report

This article has a relevant and extremely topical interest. The topic covered is of great interest to those involved in support for the neonatal transition and neonatal resuscitation.

This research does not add important and supported data for the resolution of the problem treated, but by presenting interesting examples of incomplete exhalations during positive pressure ventilation, it can be a stimulus for further research.

Introduction

The introduction chapter could be improved. In particular I would advise the authors to better explain some sentences and probably to revise their English form. for example in lines 41-44 I would divide the two concepts exposed into two separate and more in-depth sentences (RFM and PEEP uncertainities).

In line 46 sparse did you mean scarse?

explain the sentence better on lines 46-48

After the sentence in line 68, I would suggest the authors to introduce a very important concept for those who analyze the curves recorded with an RFM, namely that of leakages. Especially if the RFM is used with a face mask, during positive pressure ventilation, leakage are often present and abundant.

Methods

How do you define a leakage? which inflations are not analyzed because they present unacceptable losses? This must be explained in the methods.

The sentence at line 103-104 is not scientifically valid. The authors should explain better in this section which were the criteria for excluding a recording or part of it from the analysis.

Results

The sentence at line 112-113 should be better explained. What is eTV and ET important for? for analysis? for comprehension of the problem? to look at during resuscitation? maybe this sentence should be extended and explained with references in the introduction.

Explanations of examples:

In the figures Volumes are inspiratory TV or expiratory TV? Please specify because this is very important for interpretation of the courves.

Example 1: this explanation is clear at lines 114-116 and in fig 1 caption.

Example 2: this explanation is not completely clear. The rise of the tidal volume of the third and fourth breath of the figure could also be related to leakages? If you look at the flow courve the last two inflations have an increasing flow at the end of the inflating cycle. This could be related to leaks and could also explain higher TV recorded and lower expiratory flows that could explain incomplete expiration. Also the lower PIPs recorded in the last two inflations could be related to leaks. If interpretation is different it must be better eplained.

Example 3: the author interpretation could be correct, but they must better explain why the pressure curve is so different in shape between the first and the second phase of the recording. Maybe total gas flow was lowered? (time to peak was higher in the second phase) was it just the effect of obstructed lung or glottis closure in the first inflations Vs more opened lung in the second? Please explain. Also specify if the methods if the resuscitator system has a blender for FiO2 that keeps flows always fixed at 8 l/min or if it needs manual blending of the gases. This second option can interfere on total gas flow and thus on PEEP administered with the T-piece.

Example 4: this explanation is clear

Example 5: This example could also be related to leaks with inflating flow not returning to zero and thus making the interpretation of incomplete expiration weak. Please explain better why the authors don't relate this example to leakages.

Discussion and conclusions are clearly exposed but must be completed with more considerations about interpretation and analysis of RFM courves that are not so univocal. Pressures and flows obtained at the face mask could not be completely related to lung mechanics (compliance, resistance...) as they are affected by several factors related to device characteristics, leakages, upper airways responses such as glottis closure and digestive ways leakages. These concepts must be better emphasized in the Respiratory function monitors and limitations chapter.

Author Response

  • Thank you for the valuable comments and questions. Below are responses and our revisions.

This article has a relevant and extremely topical interest. The topic covered is of great interest to those involved in support for the neonatal transition and neonatal resuscitation.

This research does not add important and supported data for the resolution of the problem treated, but by presenting interesting examples of incomplete exhalations during positive pressure ventilation, it can be a stimulus for further research.

  • Our research intention, not stated in the manuscript, is to provide more comprehensive analysis of datasets that have already been collected. The submitted manuscript will be used as a stepping stone for funding and collaborations. It is possible to reduce the risk by not using T-piece systems for resuscitation and exploring this by us and others will continue in parallel.

Introduction

The introduction chapter could be improved. In particular I would advise the authors to better explain some sentences and probably to revise their English form. for example in lines 41-44 I would divide the two concepts exposed into two separate and more in-depth sentences (RFM and PEEP uncertainities).

  • A sentence on RFMs has been added at the end of the paragraph. The introduction sections on systems and RFMs has been revised and expanded.

In line 46 sparse did you mean scarse?

  • Sparse is preferred by the native English authors

explain the sentence better on lines 46-48

  • An introductory sentence has been added. Sentence revised and expanded.

After the sentence in line 68, I would suggest the authors to introduce a very important concept for those who analyze the curves recorded with an RFM, namely that of leakages. Especially if the RFM is used with a face mask, during positive pressure ventilation, leakage are often present and abundant.

  • A paragraph of leakage has been added. Below is a rather long discussion or answer on this difficult area.

    Leakage makes interpretation very complex and recordings with leakage was excluded for this reason. Leakage will reduce the risk of inadvertent PEEP by reducing the expiratory resistance and time constant. During PPV leakage is normally measured as difference between inspiratory and expiratory volume. However, we are looking at end expiratory flows and pressures. At this “low pressure point” in the breath cycle the leakage is likely to be low since it was possible to provide PPV with higher pressures without leakage. This was shown in an early publication in mannikins by O´Donnell et al. (2005). It has been added as a reference in the introduction and methods sections. The standard way of describing leakage is therefore not perfect when interested in end-expiratory conditions during PPV. We have not been able to find a solution to this problem and for the time being we therefore exclude leakage.

  • Level of leakages not reported (since the method of determining them is not clearly applicable): Fig 1 leakage 1%; Fig 2 leakage 17-0%; Fig 3 leakage 15-0% left figure 0% right figure; Fig 4 leakage 5-1% left figure 0% right figure; Fig 5 leakage 1-0%

Methods

How do you define a leakage? which inflations are not analyzed because they present unacceptable losses? This must be explained in the methods.

  • This has been revised and included in the methods section.

The sentence at line 103-104 is not scientifically valid. The authors should explain better in this section which were the criteria for excluding a recording or part of it from the analysis.

  • This sentence has been deleted and the limitations in discussion expanded. The provided examples were arbitrarily chosen because they were good illustrations. We have just started a structured analysis and there is no software or tool that we are aware of that can aid us. Performing this requires a few hundred hours and method development. This requires funding and is expected to be completed within 1-2 years. As far as we know this has not been attempted before.

Results

The sentence at line 112-113 should be better explained. What is eTV and ET important for? for analysis? for comprehension of the problem? to look at during resuscitation? maybe this sentence should be extended and explained with references in the introduction.

  • This results section has been expanded and clarified. It is an illustration to aid comprehension of the problem and factors. Interpreting more complex data from RFM during clinical resuscitation is probably challenging but RR, TV and leakage is not difficult and the device used including adjustments should be easy

Explanations of examples:

In the figures Volumes are inspiratory TV or expiratory TV? Please specify because this is very important for interpretation of the courves.

  • This has been added to the methods section. The volume integration was zeroed after each exhalation. Since leakage was minimal inspiratory and expiratory volumes were similar.

Example 1: this explanation is clear at lines 114-116 and in fig 1 caption.

Example 2: this explanation is not completely clear. The rise of the tidal volume of the third and fourth breath of the figure could also be related to leakages? If you look at the flow courve the last two inflations have an increasing flow at the end of the inflating cycle. This could be related to leaks and could also explain higher TV recorded and lower expiratory flows that could explain incomplete expiration. Also the lower PIPs recorded in the last two inflations could be related to leaks. If interpretation is different it must be better eplained.

  • All the recordings have very low level of leakage. The first and fourth, breath as you have noticed has a small leakage. If important it attenuates the incomplete exhalation (or at least the measurement of it). The first breath the flow at the end of exhalation not returning to zero. The fourth breath (not used an example) also has a low level of leakage but no sign of incomplete exhalation. If this is related to leakage at the end of the fourth breath, the patient having lower resistance or something else we don’t know.

Example 3: the author interpretation could be correct, but they must better explain why the pressure curve is so different in shape between the first and the second phase of the recording. Maybe total gas flow was lowered? (time to peak was higher in the second phase) was it just the effect of obstructed lung or glottis closure in the first inflations Vs more opened lung in the second? Please explain. Also specify if the methods if the resuscitator system has a blender for FiO2 that keeps flows always fixed at 8 l/min or if it needs manual blending of the gases. This second option can interfere on total gas flow and thus on PEEP administered with the T-piece.

  • This has been revised. The compliance and TV dramatically increased. With this the inspiratory rise time and interaction with resistance gives the recorded shape. The time to peak reflects the differences in inspiratory time constants but loops would have to be assessed. We have not worked with inspiratory time constants a but time to peak will be depending on fresh gas flow as well as time constants (as you suggest).

    We have added a short description to the legend. Even if the manuscript only focus on exhalation, low fresh gas flows will have this effect. The FG flow was constant, and a blender used. This has been added in the methods section.

Example 4: this explanation is clear

Example 5: This example could also be related to leaks with inflating flow not returning to zero and thus making the interpretation of incomplete expiration weak. Please explain better why the authors don't relate this example to leakages.

  • If it would have been related to leakage the flows would have been offset upwards which was not the case. This would have given a volume integration that shift upward and this was not seen. That volume still rise at the end of inspiration probably have to do with the lungs allowing more volume to enter.

Discussion and conclusions are clearly exposed but must be completed with more considerations about interpretation and analysis of RFM courves that are not so univocal. Pressures and flows obtained at the face mask could not be completely related to lung mechanics (compliance, resistance...) as they are affected by several factors related to device characteristics, leakages, upper airways responses such as glottis closure and digestive ways leakages. These concepts must be better emphasized in the Respiratory function monitors and limitations chapter.

  • This section and the clinical importance section has been revised and expanded.

Reviewer 2 Report

In this manuscript, the authors present a small case series of neonatal resuscitations in which respiratory function monitoring had been performed, and in which review of the physiologic tracings revealed evidence of incomplete exhalation.  They discuss the theoretical basis for these events, and briefly explore their clinical implications.

Overall, the paper is well written, and the abstract reflects the contents of the body text, except perhaps in failing to convey to the reader the estimated frequency of the incomplete exhalations in their study of the 129 term newborns.

In the Methods, the authors state that only high-quality recordings with no artifacts and with minimal leakage were reviewed, and that a large number of breaths and recordings were unsuitable and could not be used as examples.  This is important information, and some estimate of the relative frequency of analyzable breaths would be useful; also, only 4-second segments of monitor tracings are shown, and it is unclear whether findings within those 4 seconds persist, and if so for how long.  Persistent incomplete exhalations may have a very different significance from 3 or 4 breaths where this phenomenon is shown to occur.

In the Results, Table 1 is useful, but it could be improved if some comment on duration of incomplete exhalation was made, for each baby.

In the Discussion, on line 174, when authors state "or decreased inspiratory–expiratory ratio", do they mean “increased…”?

Also in the discussion, the authors may want to address whether the findings which suggest incomplete exhalation may be mimicked by other conditions and or artifacts, and precautions which they may have taken and that other researchers should take when attempting to study this phenomenon.

Author Response

  • Thank you for the valuable comments and questions. Below are responses and our revisions.

In this manuscript, the authors present a small case series of neonatal resuscitations in which respiratory function monitoring had been performed, and in which review of the physiologic tracings revealed evidence of incomplete exhalation.  They discuss the theoretical basis for these events, and briefly explore their clinical implications.

Overall, the paper is well written, and the abstract reflects the contents of the body text, except perhaps in failing to convey to the reader the estimated frequency of the incomplete exhalations in their study of the 129 term newborns.

  • Reviewer 1 also commented on the value of more solid analysis of the complete data set. Our research intention, not stated in the manuscript, is to continue this work with more comprehensive analysis of datasets that have already been collected.

    One of these is the Stavanger T-piece data. We also have access to other T-piece data sets and self-inflating bag data. The submitted manuscript will be used as a stepping stone for funding and collaborations. It is probably possible to reduce the risk by not using T-piece systems for resuscitation and exploring this by us and others will continue in parallel.
  •  
  • We have revised some sentences to clarify that we only screened the 129 recordings for examples to explain the concept. We have not clearly stated that this is our first analysis and that more will follow depending on funding and if we can find the right researchers to work with this rather complex project.

In the Methods, the authors state that only high-quality recordings with no artifacts and with minimal leakage were reviewed, and that a large number of breaths and recordings were unsuitable and could not be used as examples.  This is important information, and some estimate of the relative frequency of analyzable breaths would be useful; also, only 4-second segments of monitor tracings are shown, and it is unclear whether findings within those 4 seconds persist, and if so for how long.  Persistent incomplete exhalations may have a very different significance from 3 or 4 breaths where this phenomenon is shown to occur.

In the Results, Table 1 is useful, but it could be improved if some comment on duration of incomplete exhalation was made, for each baby.

  • Analysing complete recordings and change over time require new methods or extensive manual work. Variability in leakage, compliance, ventilation and spontaneous breathing efforts will be a major challenge. The work has been started but results are not available for inclusion in this manuscript. What we currently are looking at including in the next manuscript are; frequency and level of incomplete exhalation, duration of PPV and dynamics, inconclusive recordings due to leakage and poor recording quality (the users were blinded). We are not aware of any similar work published, software that can aid analysis or applicable standards.

In the Discussion, on line 174, when authors state "or decreased inspiratory–expiratory ratio", do they mean “increased…”?

  • Thank you for pointing out this error. It has been corrected. The same mistake in results section has also been corrected.

Also in the discussion, the authors may want to address whether the findings which suggest incomplete exhalation may be mimicked by other conditions and or artifacts, and precautions which they may have taken and that other researchers should take when attempting to study this phenomenon.

  • We have revised parts of the discussion. We believe that there might be more efficient ways when analysing incomplete exhalations and expiratory time constants. A technique that we do not think has been attempted is determining this using flow-volume loops.

    We have not gathered enough experienced to suggest (or compare) ways to analyse large data sets or if use during ongoing resuscitation is possible. We hope that our submitted manuscript will increase interest and collaborations.

Round 2

Reviewer 1 Report

The paper has been improved. Interest for this topic can be high.